# GRABLI: CROSS-MODAL KNOWLEDGE GRAPH ALIGNMENT FOR BIOMEDICAL LANGUAGE MODELS

## ABSTRACT

Pre-trained Language Models (LMs) have given a significant performance growth in a variety of language-related texts in biomedical domain. However, existing biomedical LLMs demonstrate a limited understanding of complex, domain-specific concept structure and the factual information stored in biomedical Knowledge Graphs (KGs). We propose **GRABLI** (Knowledge **Gra**ph and **B**iomedical Language Model A**li**gnment), a novel pre-training method that enriches an LM with external knowledge by simultaneously learning a separate KG encoder and aligning LM and graph representations. Given a textual sequence, we normalize biomedical concept mentions to the Unified Medical Language System (UMLS) KG and use the local KG subgraphs as cross-modal positive samples for mentioned concepts. Our empirical results demonstrate that applying our proposed method to various state-of-the-art biomedical LMs including PubMedBERT and BioLinkBERT, enhances their performance on diverse language understanding tasks, even after brief pre-training on a small alignment dataset derived from PubMed scientific abstracts.

## 1 INTRODUCTION

In recent years, advancements in biomedical Natural Language Processing (NLP) have been largely driven by the development of domain-specific pre-trained Language Models (LMs) (Alsentzer et al., 2019; Beltagy et al., 2019; Michalopoulos et al., 2021; Yasunaga et al., 2022b; Gu et al., 2022; Mannion et al., 2023; Sakhovskiy et al., 2024). Despite the recent success of Large Language Models (LLMs) in the general domain, they fall short of lightweight domain-specific biomedical LMs (Gu et al., 2022; Yasunaga et al., 2022b) by a large margin (Chen et al., 2023; Bai et al., 2024). While domain-specific models have shown remarkable performance on biomedical NLP benchmarks, for instance, on the Biomedical Language Understanding and Reasoning Benchmark (BLURB) Gu et al. (2022), they have been shown to impose limited domain-specific factual knowledge understanding (Sung et al., 2021; Meng et al., 2022).

The concept structure and factual knowledge within a specific domain are often represented through extensive knowledge graphs (KGs), which can describe millions of domain-specific concepts and their inter-relations. A notable example in the biomedical domain is the Unified Medical Language System (UMLS)[1] KG (Bodenreider, 2004), a comprehensive meta-thesaurus covering over 4M concept from 166 lexicons/thesauri. Recent lines of research have iteratively improved the current state-of-the-art performance on biomedical entity representations by pre-training either on UMLS concept names (Liu et al., 2021a;b; Yuan et al., 2022) or aligned text-KG subgraph pairs (Sakhovskiy et al., 2024). However, these work mostly fine-tuned LMs for entity linking, limiting their applicability beyond this specific task. This narrow focus can hinder the models' ability to generalize across diverse biomedical texts and concepts.

Recent efforts to improve the knowledge capabilities of LMs involve integrating text and knowledge graphs (KGs) in a shallow or one-way manner (Zhang et al., 2019b; Wang et al., 2021b; Sun et al., 2021; Baek et al., 2023) (e.g., from KG to text for retrieval-augmented methods like RAG (Lewis et al., 2020), REALM Guu et al. (2020), and REPLUG (Shi et al., 2024)), which could hinder multi-hop reasoning. Another approach is using an interaction token (Zhang et al., 2022; Yasunaga et al.,

---

[1]https://www.nlm.nih.gov/research/umls/index.html

2022a) or a projector Tian et al. (2024) that depends on implicit exchanges between modalities. Unlike previous efforts, we explore the alignment of the uni-modal embedding spaces using anchors to better capture interconnected information and dependencies between textual and graph modalities. This alignment may contribute to enhanced multi-hop reasoning capabilities, as the model can more effectively traverse and reason across the aligned spaces.

In this paper, we introduce Knowledge **Gra**ph and **B**iomedical Language Model A**li**gnment (GRABLI), a novel pre-training approach that enhances LM with external knowledge by concurrently training a distinct KG encoder and aligning the representations of both the LM and the graph. Specifically, as in Figure 1, given a (text, local KG) pair, a graph neural network (GNN) is utilized to capture and encode the graph knowledge into node embeddings, while pre-trained LM is used to obtain textual entity representations. Textual entity representations and concept node representations are used as anchors to align the two uni-modal embedding spaces. In this work, we seek to answer the following research questions (RQs):

**RQ1:** Is the proposed cross-modal LM-KG alignment procedure with explicit alignment between two representation spaces beneficial for biomedical NLP downstream tasks?

**RQ2:** What is the most effective graph representation for LM-KG alignment?

**RQ3:** Is the utilization of an external graph encoder more effective for cross-modal LM-KG alignment or using graph linearization followed by LM encoding is sufficient?

To comprehensively assess our model, we perform extensive experiments across several benchmarks for question answering and entity linking tasks. Initially, we pretrain several LMs with GRABLI, leveraging the PubMed corpus and UMLS KG. Our experiments demonstrate that GRABLI outperforms several biomedical language models, including BioLinkBERT (Yasunaga et al., 2022b) and PubMedBERT (Gu et al., 2022). Specifically, PubMedBERT shows mean accuracy improvements of 2.1%, 1.7%, and 6.2% on the PubMedQA, MedQA, and BioASQ benchmarks, respectively. GRABLI significantly enhances the ability of LMs to generate distinguishable and informative representations of biomedical concepts. In particular, BioLinkBERT with GRABLI pretraining performs on par or slightly better than the task-specific SapBERT model, which is pre-trained on 12M UMLS triples (4M concept nodes). Our research highlights that our cross-modal knowledge graph alignment, applied to both text and the knowledge graph, notably enhances language-knowledge representations after a small pre-trainning stage involving 1.5M sentences and 600K nodes only. The source code as well as pre-trained models will be released upon paper acceptance.

## 2 RELATED WORK

**Knowledge-Augmented Language Models** One line of research on knowledge-enhanced LMs (Liu et al., 2020; Sun et al., 2020; Ke et al., 2021; Mannion et al., 2023; Yuan et al., 2022; Moiseev et al., 2022) attempted to infuse factual information into LM input either by augmenting natural language texts with relational triples or directly training on relational triples. Various methods (Zhang et al., 2019a; He et al., 2020; Wang et al., 2021a; Peters et al., 2019; Rosset et al., 2020; Yu et al., 2022; Kang et al., 2022) augment in-context entity representation with external knowledge retrieved from KG. While such methods are able to improve quality on NLP tasks, they usually perform unidirectional information fusion for improved LM embeddings using either a single LM for both modalities or static KG node embeddings. Static node embeddings are unable to capture node semantics and only capture structural information, Transformer-based (Vaswani et al., 2017) LM's architecture is inherently dense which confronts the sparse nature of KGs. Recently proposed GreaseLM (Zhang et al., 2022) and DRAGON (Yasunaga et al., 2022a) models improve LM reasoning ability by introducing bidirectional cross-modal interaction text and grounded KG subgraph interaction through specialized cross-modal LM token for enhanced question answering. However, both models depend on implicit intermodal exchanges: the LM accesses KG information via a single token initialized with pooled subgraph representation, while the graph encoder receives semantic input through an interaction node initialized with pooled sentence representation. Meanwhile, these modalities offer complementary representations of a single entity, capturing different contexts: sentences for the LM and KG subgraphs for the graph encoder implying that the two uni-modal spaces can be aligned through entities serving as anchors in a unified embedding space.

Recently, Tian et al. (2024) proposed a method that encodes subgraphs based on the entities present in the question and options. In contrast with direct feeding of KG triples into LLMs Baek et al. (2023), this approach utilizes a GNN, a cross-modality pooling module, and a domain projector to send the encoded subgraphs to LLMs for inference, alongside the input text embeddings. This represents an alternative prompt-based direction focusing on parameter-efficient fine-tuning.

**Graph Representation Learning** A series of translation-based node representation methods (Yang et al., 2015; Bordes et al., 2013; Trouillon et al., 2016; Kazemi & Poole, 2018; Sun et al., 2019) models a relation triplet (graph edge) as a translation between head and tail nodes. Initially, these methods learned static node embedding matrix as well as relation embeddings via the link prediction task contrastively with knowledge triples present in a KG being positive samples and non-present ones being negative samples. Experimental evaluation of translation-based methods for biomedical concept representation (Chang et al., 2020) indicates that these methods fall short of the LM-based approach due to a lack of essential semantical information present in texts. While translation-based methods model each edge individually, Message Passing (MP) (Gilmer et al., 2017) graph neural networks obtain node embeddings by passing and aggregating messages from multiple neighboring nodes at once. Various architectures under the MP framework mostly differ in message aggregation function. For instance, GraphSAGE (Hamilton et al., 2017) performs mean-pooling over neighboring nodes, and Graph Attention Network (GAT) (Velickovic et al., 2018; Brody et al., 2022) applies an attention-based aggregation. In our work, we adopt GAT for local KG subgraph aggregation as it has proved itself an effective graph encoder for LM-KG interaction applications (Yasunaga et al., 2021; Zhang et al., 2022; Yasunaga et al., 2022a; Sakhovskiy et al., 2023; 2024). Another approach (Wang et al., 2021b; Salnikov et al., 2023) gets rid of additional memory footprint introduced by an external graph encoder by linearizing KG subgraphs into textual strings encoded with an LM.

**Cross-Modal Alignment** Our research is inspired by recent advancements in aligning multiple uni-modal representations across various domains. Koh et al. (2023a;b) trains a small alignment network to align images with their captions for cross-modal visual and textual generative tasks. Liu et al. (2023) learns a lightweight projection to align visual and textual features for improved multimodal image and language understanding. Ke et al. (2021) introduced a method to align entities in text with their representations in graphs, enhancing graph summarization. Unlike prior work, we perform explicit cross-modal alignment by directly minimizing distances between cross-modal paired representations of a single biomedical concept.

## 3 PROBLEM STATEMENT/NOTATION

**Biomedical Knowledge Graph** Formally, a Knowledge Graph can be defined as $\mathcal{G} = (\mathcal{V}, \mathcal{E}, \mathcal{R})$, where $\mathcal{V}$ is the set of biomedical concepts, $\mathcal{E} \subset \mathcal{V} \times \mathcal{R} \times \mathcal{V}$ is the set of labeled edges, and $\mathcal{R}$ are possible relation types. In UMLS, one of the largest biomedical KGs, a node $v \in \mathcal{V}$ can be represented with a set of $k \geq 1$ distinct synonymous concept names $S_v = \{s_1^v, s_2^v, \ldots, s_k^v\}$. Thus, a concept $v \in \mathcal{V}$ can be represented with two complementary modalities: (i) a textual modality described by $S_v$, (ii) a KG modality expressed with local subgraph $\mathcal{G}_v \subset \mathcal{G}$ centered around $v$. Additionally, textual concept representations can be learnt from raw texts they are mentioned in.

**KG Subgraphs** From KG perspective, a node $v \in \mathcal{V}$ can be described by the structure of its local KG subgraph, denoted as $\mathcal{G}_v = (\mathcal{V}_v, \mathcal{E}_v, \mathcal{R}) \subset \mathcal{G}$, consisting of 1-hop neighbors subgraph centered around $v$: $\mathcal{E}_v = \{(u, r, v) \in \mathcal{E}\}, \mathcal{V}_v = \{u \mid (u, r, v) \in \mathcal{E}_v\} \cup v$. Here, $\mathcal{G}_v$ can be viewed as a structural KG-induced context for a concept. Following Hamilton et al. (2017), we sample a subset of up to 3 neighboring nodes to reduce computational cost of our model.

**Alignment Intuition** While graph encoder $GNN$ can capture the hierarchy of in-domain concepts along with other inter-concept relationships, textual encoder $LM$ can provide deeper insights into concept semantics learnt from raw texts. Conversely, $LM$ may struggle to effectively learn the intricate concept structure from texts alone. Thus, we assume that two embedded representations $\bar{g}_v$ and $\bar{e}_v$ are complementary representations encoding different features of the same concept $v$. Our goal is to align these two uni-modal entity representations by enabling a mutual knowledge exchange. Since we assume $\bar{e}_v$ and $\bar{g}_v$ to be complementary representations capturing different features of a concept

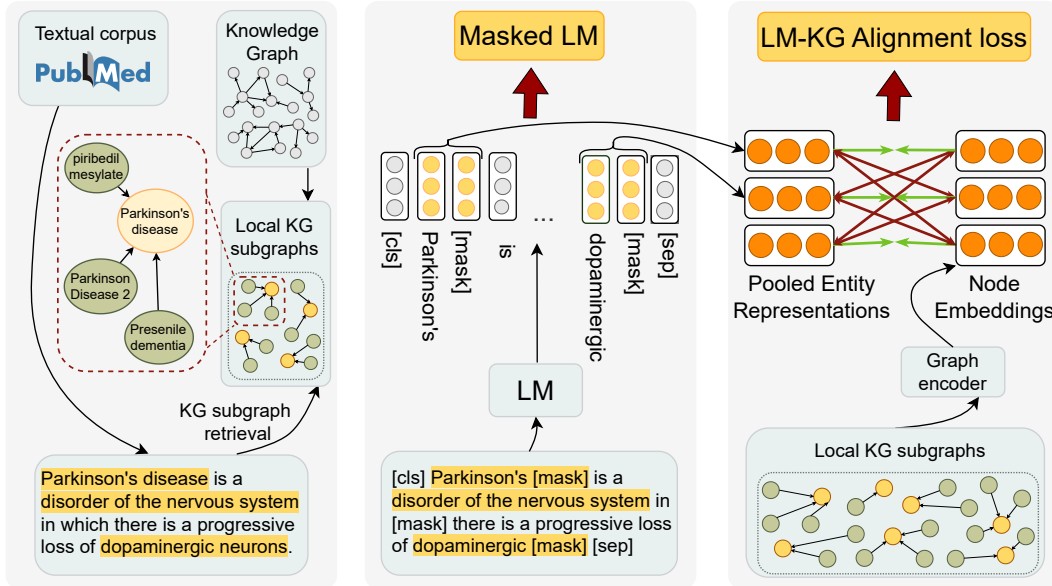

Figure 1: The overall framework. We first retrieve subgraphs from the knowledge graph based on the entities in a text fragment (§3). We then develop **GRABLI** (Knowledge **Gra**ph and **B**iomedical Language Model A**li**gnment) to align knowledge between a textual encoder and a graph encoder (§4.1). We utilize two objectives: (1) masked language modeling (MLM), which masks some tokens in the input text and then predicts them, and (2) cross-modal alignment, which pull two representations of a concept closer in a combined embedding space. Since the entity representation is pooled over a textual sequence masked for MLM objective, this alignment objective further enforces LM to infer relevant information from the whole sequence (§4.2).

$v$, we propose to use these embeddings as anchors for aligning inner representations of $GNN$ and $LM$.

## 4 METHODOLOGY

Overall, our objective is to align the knowledge between a textual encoder and a graph encoder using textual entity representations and concept node representations as anchors for aligning two uni-modal embedding spaces.

### 4.1 UNI-MODAL REPRESENTATIONS

**Entity Representations** Let $T = (t_1, t_2, \ldots, t_N)$ denote a textual sequence consisting of $N$ tokens. To encode the sequence, we adopt a language model $LM$ that is based on Transformer encoder (Vaswani et al., 2017):

$$H_T = (\bar{h}_1, \bar{h}_2, \ldots, \bar{h}_N) = LM\{(t_1, t_2, \ldots, t_N)\},$$

where $\bar{h}_j \in \mathbb{R}^d$ is a $d$-dimensional embedding for $j$-th token in the sequence. Here, $H_T \in \mathbb{R}^{N \times d}$ is a textual embedding matrix for a sequence $T$. We assume that the text $T$ mentions $M$ KG nodes, denoted as $V_T = \{v_i\}_{i=1}^M \subset \mathcal{V}$. For each concept $v \in \mathcal{V}_T$ there is a subset of tokens from $T$ corresponding to it with respective embeddings $H_v \subset H_T$. A pooled entity representation $\bar{e}_v \in \mathbb{R}^d$, contextualized by sequence $T$, is computed as the mean of token embeddings $H_v$:

$$\bar{e}_v = \frac{1}{|H_v|} \sum_{\bar{h}_j \in H_v} \bar{h}_j$$

**Subgraph Node Representations**    A $d$-dimensional graph-based representation $\bar{g}_v \in \mathbb{R}^d$ for concept $v$ can be obtained by encoding local KG subgraph $\mathcal{G}_v$ with a graph encoder: $\bar{g}_v = GNN(\mathcal{G}_v)$. To obtain a KG-based vector representation $\bar{g}_v$ for $v$, we utilize a multi-layer Graph Attention Network (GAT) (Velickovic et al., 2018; Brody et al., 2022) that iteratively updates node representation under Message Passing framework (Gilmer et al., 2017):

$$\bar{g}_v^{(l)} = \sigma \left( \sum_{(u,r,v) \in \mathcal{E}_v} \alpha_{uv}^l \cdot W^l \bar{g}_u^{(l-1)} + W_o^l \bar{g}_v^{(l-1)} \right)$$

$$\alpha_{uv}^l = \frac{exp(e_{uv}^l)}{\sum_{(w,r,v) \in \mathcal{E}_v} exp(e_{wv}^l)} \qquad e_{uv}^l = a^T \cdot \sigma(W^l \cdot [\bar{g}_u^{(l-1)} \parallel \bar{g}_v^{(l-1)}]),$$

where $\alpha_{uv}$ is the attention weight for an edge $(u, r, v)$, $W^l, W_o^l \in \mathbb{R}^{d \times d}$ are weight matrices, $l$ is a layer number, and $\sigma$ is a LeakyRELU activation. As an initial representation for a node $u$, a random concept name $s_u \in S_u$ is sampled and encoded with a textual encoder: $\bar{g}_u^{(0)} = LM(s_u)$. Thus, graph encoder $GNN$ is provided with additional semantics captured by the textual encoder $LM$.

**Linearized Graph Representation**    An alternative to the introduction of an additional external graph neural network is to linearize a set of graph triples into a textual graph summary encoded with an LM (Liu et al., 2020; Ke et al., 2021; Salnikov et al., 2023). Since KG nodes are often attributed with textual representations (e.g., textual concept names in UMLS KG), this approach allows transferring knowledge learned from raw texts to graph representations. To obtain a linearization $L(\mathcal{G}_v)$ of graph $\mathcal{G}_v$, we linearize each edge $(u, r, v) \in \mathcal{E}_v$ as: "$L_{(u,r,v)} = s_u\ r\ s_v\ [SEP]$", where $s_u \in S_u$ is a randomly sampled name of concept $u$. The resulting linearized graph obtained as the concatenation of concept name $s_v \in S_v$ and linearized edges from $\mathcal{E}_v$ is further encoded with a textual encoder:

$$\bar{g}_v = LM \left( [CLS]\ s_v\ [SEP] \bigoplus_{(u,r,v) \in \mathcal{E}_v} L(u, r, v) \right),$$

where $\bigoplus$ is a string concatenation.

### 4.2    TRAINING OBJECTIVES

**Masked Language Modeling (MLM)**    MLM, a widely used pretraining objective for language models, has proven effective both in the general domain Devlin et al. (2019); Liu et al. (2019); Yasunaga et al. (2022a) and in the biomedical domain Gu et al. (2022); Yasunaga et al. (2022b;a). The objective aims to make a model learn informative token representations $H_T$ by predicting masked tokens from unmasked ones using a corrupted input text as context. Specifically, given a subset of tokens $M \subset T$ masked with a masking token $[MASK]$, the model aims to restore the original tokens relying on the remaining ones as context:

$$\mathcal{L}_{MLM} = - \sum_{t_i \in M} \log p(t_i | H_T)$$

**Cross-Modal Alignment**    Our alignment procedure is designed to enhance a textual encoder $LM$ with domain-specific knowledge through contrastive learning using mentioned entities as anchors. Specifically, given a batch $\{(\bar{e}_i, \bar{g}_i)\}_{i=1}^B$ consisting of $B$ aligned paired text-graph representations, we introduce a InfoNCE (van den Oord et al., 2018) contrastive objective to pull two representations a biomedical concept $v_i$ closer in the aligned embedding space:

$$\mathcal{L}_{align} = -\frac{1}{B} \sum_{i=1}^B \left( \log \frac{\exp(cos(\bar{e}_i, \bar{g}_i)/\tau)}{\sum_{j=1}^B \exp(cos(\bar{e}_j, \bar{g}_j)/\tau)} \right),$$

where $B$ is a batch size, and $\tau > 0$ is a temperature parameter, and $cos(\bar{e}_i, \bar{g}_i)$ is a cosine similarity between $\bar{e}_i$ and $\bar{g}_i$. Since the entity representation $\bar{e}_i$ is pooled over a textual sequence masked for

Table 1: Mean evaluation accuracy and standard deviation across 10 evaluation runs for proposed GRABLI alignment procedure on biomedical question answering datasets. *GNN* stands for GRABLI with external GAT grah encoder while *Linear graph* stands for single-encoder implementation with KG subgraphs encoded with an LM.

| Model | PubMedQA | MedQA | BioASQ 2023 |
|---|---|---|---|
| PubMedBERT | $63.1 \pm 2.9$ | 38.1 | $67.8 \pm 4.1$ |
| + GRABLI (GNN) | $65.2 \pm 1.2$ | 39.8 | $74 \pm 3.4$ |
| + GRABLI (Linear graph) | $65.0 \pm 1.6$ | 38.81 | $72.2 \pm 4.4$ |
| BioLinkBERT$_{base}$ | $63.3 \pm 3.6$ | 40.0 | $65.9 \pm 2.7$ |
| + GRABLI (GNN) | $64.4 \pm 2.1$ | 43.1 | $73.6 \pm 3.6$ |
| + GRABLI (Linear graph) | $63.86 \pm 4.4$ | 40.46 | $65.70 \pm 3.6$ |
| BioLinkBERT$_{large}$ | $69.52 \pm 2.4$ | 44.6 | $67.7 \pm 3.7$ |
| + GRABLI (GNN) | $68.72 \pm 5.2$ | 45.01 | $67.91 \pm 4.5$ |
| + GRABLI (Linear graph) | $70.9 \pm 1.7$ | 45.5 | $66.0 \pm 6.1$ |
| **Task-specific joint LM-KG reasoning QA methods** | | | |
| QA-GNN (Yasunaga et al., 2021) | 72.1 | 45.0 | — |
| GreaseLM (Zhang et al., 2022) | 72.4 | 45.1 | — |
| DRAGON (Yasunaga et al., 2022a) | 73.4 | 47.5 | — |

MLM objective, alignment loss further enforces LM to infer relevant information from the whole sequence $T$.

The resulting loss is a sum of MLM and alignment objective: $\mathcal{L} = \mathcal{L}_{MLM} + \mathcal{L}_{align}$. Intuitively, the training objective is designed to encourage an LM enrich entity representation with external knowledge from a KG while retaining its language understanding through continious MLM pre-training.

## 5 EXPERIMENTS

To assess the effectiveness of our proposed methodology, we first pre-train existing biomedical LMs with the GRABLI method and then assess the performance of the resulting models in various biomedical NLP tasks.

**Pretraining Data** As pretraining data, we adopt the PubMed abstracts[2] with biomedical entities recognized and normalized to the UMLS KG (version 2020AB) with the BERN2 tool (Sung et al., 2022). Given the substantial entity distribution imbalance in scientific abstracts, with entities like human, mice, and cancer being the most common ones, we address this issue as follows. To ensure a more balanced dataset with diverse concepts, we sample only up to 10 sentences from PubMed abstracts iteratively for each concept present in the UMLS. The resulting dataset has 1.67M sentences with mentioned entities covering about 600K unique UMLS concepts.

### 5.1 EVALUATION TASKS

We evaluate the effectiveness of our proposed alignment method on the following knowledge-demanding tasks in biomedical domain:

**Question Answering (QA)** For our experiments, we adopt three QA datasets: (i) PubMedQA (Jin et al., 2019); (ii) MedQA-USMLE (Jin et al., 2021); (iii) BioASQ 2023 (Nentidis et al., 2023). PubMedQA is a dataset containing 1,000 questions derived from PubMed abstracts, with each question having a single correct answer chosen from yes/no/maybe options. MedQA-USMLE is the collection of 12,723 multiple-choice questions derived from the US National Medical Board Examination, each offering 4 answer choices. BioASQ includes 1,357 binary yes/no questions manually curated by experts in the biomedical domain.

---

[2]pubmed.ncbi.nlm.nih.gov/

Table 2: Evaluation results on biomedical entity linking in zero-shot and supervised set-ups. @1 and @5 stand for Accuracy@1 and Accuracy@5, respectively. For each model, underline highlights the best of two scores: (i) retrieval accuracy of the original biomedical LM and (ii) the score for model pre-trained with GRABLI method.

| Model | NCBI | | BC5CDR-D | | BC5CDR-C | | BC2GM | | SMM4H | |
|---|---|---|---|---|---|---|---|---|---|---|
| | @1 | @5 | @1 | @5 | @1 | @5 | @1 | @5 | @1 | @5 |
| **Zero-shot evaluation** | | | | | | | | | | |
| PubMedBERT | 49.51 | 65.69 | 58.75 | 75.04 | 76.24 | 80.24 | 68.12 | 74.11 | 16.13 | 25.27 |
| + GRABLI | 68.14 | 79.90 | 72.30 | 81.28 | 85.65 | 89.65 | 83.25 | 89.44 | 24.91 | 36.82 |
| BioLinkBERT$_{base}$ | 35.78 | 44.12 | 45.81 | 54.64 | 70.59 | 73.41 | 58.17 | 61.52 | 8.30 | 10.83 |
| + GRABLI | 68.63 | 78.92 | 73.82 | 82.65 | 86.59 | 90.82 | 82.64 | 89.24 | 27.92 | 43.08 |
| BioLinkBERT$_{large}$ | 32.35 | 42.65 | 44.29 | 50.99 | 70.12 | 73.18 | 57.66 | 62.13 | 8.54 | 12.27 |
| + GRABLI | 70.1 | 78.92 | 73.21 | 80.67 | 85.65 | 90.12 | 82.44 | 89.04 | 22.98 | 34.78 |
| SapBERT | 71.57 | 84.31 | 73.67 | 84.32 | 85.88 | 91.29 | 87.61 | 92.18 | 39.59 | 58.84 |
| + GRABLI | 71.57 | 81.86 | 74.28 | 82.50 | 86.35 | 90.35 | 85.89 | 91.37 | 28.04 | 42.00 |
| GEBERT | 70.59 | 83.33 | 74.58 | 85.39 | 85.41 | 91.76 | 87.21 | 92.79 | 38.27 | 62.33 |
| + GRABLI | 73.04 | 81.86 | 73.52 | 82.50 | 86.59 | 92.0 | 85.48 | 91.57 | 28.04 | 46.21 |
| **Supervised evaluation** | | | | | | | | | | |
| PubMedBERT | 72.06 | 84.31 | 74.73 | 83.71 | 86.12 | 92.00 | 87.92 | 92.39 | 66.19 | 79.90 |
| + GRABLI | 74.02 | 82.35 | 74.73 | 81.74 | 87.76 | 92.94 | 88.32 | 91.88 | 68.71 | 79.66 |
| BioLinkBERT$_{base}$ | 56.86 | 70.59 | 74.58 | 85.39 | 87.29 | 92.94 | 88.32 | 92.39 | 65.94 | 77.74 |
| + GRABLI | 75.00 | 84.31 | 75.49 | 83.26 | 88.94 | 92.71 | 88.32 | 92.89 | 67.27 | 78.34 |
| SapBERT | 75.00 | 85.78 | 74.58 | 84.47 | 86.59 | 93.18 | 89.24 | 93.71 | 66.79 | 80.51 |
| + GRABLI | 74.51 | 83.82 | 74.73 | 82.80 | 88.24 | 93.18 | 88.12 | 92.79 | 69.19 | 78.94 |
| GEBERT | 73.04 | 84.80 | 75.80 | 85.39 | 87.06 | 92.71 | 88.83 | 93.71 | 65.70 | 80.63 |
| + GRABLI | 74.02 | 83.33 | 75.49 | 83.87 | 89.41 | 93.65 | 88.22 | 93.50 | 67.51 | 80.75 |

**Entity Linking (EL)** For biomedical entity linking, we adopt 5 corpora: (i) NCBI Dogan et al. (2014), (ii) BC5CDR-D Li et al. (2016), (iii) BC5CDR-D Li et al. (2016), (iv) BC2GN Morgan et al. (2008), (v) SMM4H Sarker et al. (2018). We consider two scenarios: (i) zero-shot similarity-based retrieval approach over pooled mention and concept name representations (Tutubalina et al., 2020a); (ii) supervised approach based on BioSyn (Sung et al., 2020), a model that iteratively updates candidates list using synonym marginalization. Following prior EL research (Phan et al., 2019; Sung et al., 2020; Tutubalina et al., 2020a; Sakhovskiy et al., 2024), we employ the top-$k$ accuracy as the evaluation metric: $\text{Acc@k} = 1$ if the correct concept is retrieved at the rank $\leq k$, otherwise $\text{Acc@k} = 0$. For more details on adopted datasets as well as evaluation details please see Appendix C.

**Relation Extraction** Additionally, we perform evaluation on three biomedical relation extraction datasets: (i) Chemical Protein Interaction corpus (ChemProt) (Krallinger et al., 2017), (ii) Drug-Drug Interaction corpus (DDI) (Herrero-Zazo et al., 2013), and (iii) Genetic Association Database (GAR) (Bravo et al., 2015). For evaluation results, see Appendix B.

**Pre-training set-up & Implementation Details.** We trained our models for 65k steps (10 epochs) with a batch size of 256 using AdamW (Loshchilov & Hutter, 2019) optimizer with a peak learning rate of $2 \cdot 10^{-5}$ for LM parameters and $1 \cdot 10^{-4}$ for other parameters and cosine learning rate decay to zero. For MLM objective, we follow the original set-up proposed in BERT (Devlin et al., 2019) by selecting 15% of input tokens. Each selected token is either replaced with a special $[MASK]$ token, left unchanged, or replaced by a randomly selected vocabulary token with probabilities of 0.8, 0.1, and 0.1, respectively. As base models, we adopt PubMedBERT[3] (Gu et al., 2022) and BioLinkBERT[4][5] (Yasunaga et al., 2022b), state-of-the-art biomedical LMs that are pre-trained on sci-

---

[3]huggingface.co/microsoft/BiomedNLP-BiomedBERT-base-uncased-abstract-fulltext
[4]huggingface.co/michiyasunaga/BioLinkBERT-base
[5]huggingface.co/michiyasunaga/BioLinkBERT-large

entific articles from PubMed. In our experiments, we pre-train each *base-* and *large-*sized GRABLI model for 65K with a batch size of 256. For more details please see Appendix A.

**Evaluation set-up** To explore the effectiveness of GRABLI, we compare each pre-trained alignment model against its base model with the original weights. Notably, PubMedBERT and BioLinkBERT models are also trained on scientific texts from PubMed database and only differ in pre-training objective. Additionally, we employ task-specific QA-GNN (Yasunaga et al., 2021) and GreaseLM (Zhang et al., 2022) models that enhance backbone BioLinkBERT$_{large}$ with relevant UMLS KG subgraph as well as reasoning module available during inference time. For entity linking, we adopt SapBERT (Liu et al., 2021a) and GEBERT[6] (Sakhovskiy et al., 2023) which are a PubMedBERT additionally pre-trained for synonymous concept name clusterization objective on all concepts available in the UMLS KG. GEBERT additionally performs concept clusterization in node representation space followed by representation alignment between textual and graph encoders. Due to small dataset sizes and fine-tuning instability, we average performance across 10 runs on PubMedQA and BioASQ corpora.

Table 3: Ablation analysis for GRABLI model with PubMedBERT base model. For each ablation-set-up and dataset, mean accuracy across 10 runs with different random states are reported.

| Node representation | PubMedQA | BioASQ |
|---|---|---|
| GNN (GAT) | 65.2 | **74** |
| GNN (GraphSAGE) | 58.8 | 70 |
| LM + Linear graph | 65.0 | 72.2 |
| LM + DistMult | **65.44** | 69.77 |
| LM + TransE | 64.22 | 70.47 |
| Textual | 58.86 | 71.40 |

Table 4: Ablation analysis for GRABLI model with PubMedBERT base model and GAT graph encoder. For each ablation-set-up and dataset, mean accuracy across 10 runs with different random states are reported.

| Model | PubMedQA | BioASQ |
|---|---|---|
| PubMedBERT | 63.1 | 67.8 |
| + GRABLI | 65.2 | 74 |
| **Training objective** | | |
| $-\mathcal{L}_{MLM}$ | 60.54 | 64.53 |
| $-\mathcal{L}_{align}$ | 63.78 | 70.58 |
| **Token-entity aggregation** | | |
| Weighted | 63.20 | 70.58 |
| GAT | 64.50 | 70.58 |
| Transformer layer | 63.06 | 71.40 |
| **# Graph encoder layers** | | |
| $L = 3$ | 64.72 | 71.51 |
| $L = 7$ | 62.62 | 69.07 |

## 5.2 RESULTS

To answer the **RQ 1**, we assess our methodology on biomedical QA and entity linking. The evaluation results for pre-trained GRABLI models on biomedical QA datasets are presented in Table 1. Across all datasets, GRABLI consistently boosts baseline models, for instance, PubMedBERT aligned through an external GAT encoder demonstrates 2.1%, 1.7%, and 6.2% mean accuracy gain on PubMedQA, MedQA, BioASQ, respectively.

Despite BioLinkBERT$_{large}$ has no access to a retrieved KG subgraph for inference-time reasoning, after GRABLI pretraining it performs on par or better than the task-specific QA-GNN and GreaseLM methods that reason over retrieved KG subgraphs. We note that both QA-GNN and GreaseLM have BioLinkBERT$_{large}$ as backbone LM.

Table 2 presents the evaluation results for aligned models on the QA task. As seen from the results, GRABLI increases entity linking capabilities of general-purpose biomedical LMs, especially in zero-shot settings. For instance, PubMedBERT and BioLinkBERT$_{base}$ show huge average Accuracy@1 gains of 13.1% and 24.2% across all datasets in zero-shot evaluation, respectively.

Thus, GRABLI pretraining enhances LM's ability to produce distinguishable and informative biomedical concept representations. Interestingly, BioLinkBERT$_{base}$ with GRABLI pretraining performs on par or slightly better than the task-specific SapBERT model that is pretrained on all synonyms available in UMLS on 2 of 5 corpora (namely, BC5CDR-Disease and BC5CDR-Chem). Moreover, GRABLI gives a 2.4% Accuracy@1 improvement for SapBERT in supervised set-up on SMM4H corpus.

---

[6]huggingface.co/andorei/gebert_eng_gat/

As shown in Appendix B, GRABLI achieves a marginal micro F1 score increase on all three relation extraction datasets for PubMedBERT and increases BioLinkBERT performance on 2 of 3 datasets.

## 5.3 NODE REPRESENTATION STUDY

To answer the **RQ 2** and **RQ 3**, we implement GRABLI with different graph representation methods. Under the GNN approach, we pre-train and evaluate GRABLI implementation with Graph-SAGE (Hamilton et al., 2017) instead of GAT which adopts mean-pooling instead of attention aggregation across neighboring nodes.

**Translation-based Node Representations**  In a series of graph representation methods (Yang et al., 2015; Bordes et al., 2013; Trouillon et al., 2016; Sun et al., 2019), a relation triplet (graph edge) $(u, r, v) \in \mathcal{E}$ is modeled as a relation-based translation of the head node $v$ with a relational transformation $f_r$: $u \approx f_r(v)$. In our work, we adopt DistMult Yang et al. (2015) and TransE Bordes et al. (2013) to represent in-context entity representation $\bar{e}_v$ as a transformation of concept name embedding $\bar{g}_u = f_r(LM(s_u))$ for $s_u \in S_u$.

**Textual Node Representations**  To assess the necessity of a graph encoder for capturing additional information not accessible to the language encoder LM, we perform experiments using node embeddings that rely exclusively on textual concept names. In particular, we compute a node embedding $\bar{g}_u$ by mean pooling the textual output of a randomly chosen concept name $s_u \in S_u : \bar{g}_u = LM(s_u)$.

**Analysis: Node Representation choice**  Experiments with different node representation types are summarized in Table 3. Based on the results, we can make the following observations. First, simpler mean-pooling local subgraph aggregation under the GNN-based approach leads to a significant performance drop of 6.4% and 4% on PubMedQA and BioASQ which highlights the importance of learning relative node importance scores: not all nodes are equally useful. Despite its simplicity, translation-based DistMult and TransE models show high performance in our alignment procedure in combination with LM. Similarly, a linearized graph encoded with LM seems to be the closest to GRABLI implementation with a GAT encoder. Thus, we conclude that LMs can serve as an effective graph representation method for text-attributed graph for LM-KG alignment. Finally, textual node representations with no KG subgraph provided have shown poor performance indicating the performance of additional local graph context for text-graph alignment.

## 5.4 ABLATION STUDY

To justify modeling choices made in GRABLI model, we perform an extensive analysis in three directions: (i) Training loss choice, (ii) Token-entity aggregation, (iii) graph encoder size. As token-entity aggregation method we experiment with following set-ups: (i) *weighted* aggregation which attention weights to sum token embeddings of the last LM layer with no additional transformations; (ii) *GAT* aggregation adopts single GAT layer as described in Section 4.1; (iii) Transformer layer over tokens to correspond to the same entity only. For each ablation, we pre-train a separate GRABLI model with PubMed initialization and summarize evaluation results across 10 runs with different random states on PubMedQA and BioASQ. The results are summarized in Table 4. A removal of each of two losses drops the QA quality indicating that performing token-level LM-KG alignment only leads to the degradation to LM's language understanding. Lower/higher GAT layer count as well as more complex token-entity aggregation functions do not lead to performance improvement.

## 6 CONCLUSION

We propose GRABLI, a novel self-supervised pretraining method for Knowledge Graph (KG) and Language Model (LM) alignment. Experimental results indicate that the alignment of biomedical LMs enhances their performance on both question answering and entity linking tasks in the biomedical domain after a short pre-training on 1.7M sentences only. Comparison of various graph representation methods has revealed the effectiveness of both LM-based approaches with linearized graphs as well as sparse graph neural networks for capturing vital KG context absent in raw texts. For future work, we aim to expand and apply our pre-training method to general domains and other LM architectures, such as decoder-only and encoder-decoder models.

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

## A  HYPERPARAMETER DETAILS

Table 5 lists hyperparameter values used for pre-training of GRABLI models.

For fine-tuning on QA and relation extraction experiments, we adopt the hyperparameters from BioLinkBERT (Yasunaga et al., 2022b) for better comparability of the experimental results. The main difference is that we load the model weights from the best epoch in terms of dev set quality metric. For zero-shot linking, we adopt the retrieval code from Tutubalina et al. (2020b). For BioSyn (Sung et al., 2020), we adopt the default hyperparameters.

Table 5: Hyperparameter values used for GRABLI pretraining

| Hyperparameter | Base models | Large models |
|---|---|---|
| Graph encoder hidden size | 768 | 768 |
| Max number of node neighbors | 3 | 3 |
| Number of graph encoder layers | 5 | 5 |
| GAT's number of attention heads | 2 | 2 |
| LM parameters learning rate | $2 \cdot 10^{-5}$ | $1 \cdot 10^{-5}$ |
| Non-LM parameters learning rate | $1 \cdot 10^{-4}$ | $1 \cdot 10^{-4}$ |
| Batch size | 256 | 256 |
| # of epochs | 10 | 10 |

## B  RELATION EXTRACTION

(Krallinger et al., 2017) (Herrero-Zazo et al., 2013) (Bravo et al., 2015) To assess the GRABLI pretraining effectiveness on biomedical relation extraction task, we adopt three corpora: (i) ChemProt (Krallinger et al., 2017), (ii) DDI (Herrero-Zazo et al., 2013), (iii) GAD (Bravo et al., 2015). The evaluation results are presented in Table 6. Due to computational instability of the results, we do not report the evaluation results for BioLinkBERT$_{large}$. On average, GRABLI demonstrates a marginal improvement depending on base LM and dataset.

## C  DATASETS

*The NCBI Disease Corpus* Dogan et al. (2014) contains 793 PubMed abstracts with disease mentions and their concepts corresponding to the MEDIC dictionary (Davis et al., 2012). It has 5134, 787, and 204 entities in train, dev, and test set after filtration of simple cases such as train-test and dictionary-test set intersection, respectively.

*BC5CDR* (Li et al., 2016) provides a task for the extraction of chemical-disease relations (CDR) from 1500 PubMed abstracts that contains annotations of both chemical/diseases. The disease part has 4182, 4244, and 657 entities in train, dev, and test set after filtration, respectively. The chemical part contains 5203, 5347, and 425 entities, respectively.

*BioCreative II GN* (Morgan et al., 2008) contains PubMed abstracts with human gene and gene product mentions for gene normalization (GN) to Entrez Gene identifiers (Maglott et al., 2007). There are 2,725/985 train/test entities.

The Social Media Mining for Health (*SMM4H*) challenge (Sarker et al., 2018) released a dataset with annotated ADR mentions linked to MedDRA. Tweets were collected using 250 generic and trade names for therapeutic drugs. Manually extracted ADR expressions were mapped to Preferred Terms (PTs) of the MedDRA dictionary. The dataset provides 6650/831 train/test entities.

The Chemical Protein Interaction corpus (*ChemProt*) (Krallinger et al., 2017) covers chemical-protein interactions between chemical and protein entities extracted from PubMed abstracts. In total, there are 23 interaction types. The dataset includes 18035/11268/15745 samples in train/dev/test sets.

*DDI* (Herrero-Zazo et al., 2013) is a Drug-Drug Interaction corpus designed for research on pharmaceutical information extraction. It consists sentence-level annotations for drug-drug interactions from PubMed abstracts. The corpus has 25296/2496/5716 train/dev/test samples.

*GAD* is the semi-automatically collected Genetic Association Database corpus of gene-disease interactions from PubMed abstracts. It has 4261/535/534 samples in train/dev/test.

Table 6: Evaluation results on biomedical relation extraction in terms of Micro F1. For each model, underline highlights the best quality among the original biomedical LM and model pre-trained with GRABLI method. The best results for each dataset are highilghted in **bold**.

| Model | ChemProt | DDI | GAD |
|---|---|---|---|
| PubMedBERT | 76.57 | 79.02 | 83.36 |
| + GRABLI | 76.91 | **81.17** | **83.68** |
| BioLinkBERT$_{base}$ | 76.97 | 79.79 | 81.97 |
| + GRABLI | **77.52** | 79.69 | 82.71 |

## D  HARDWARE & SOFTWARE SET-UP

All models in our experiments were trained and evaluated using the version 1.11.0 of PyTorch Paszke et al. (2019) with CUDA 11.3 Nickolls et al. (2008) support. GAT (Brody et al., 2022) and Graph-SAGE (Hamilton et al., 2017) graph neural networks were adopted from the PyTorch Geometric Fey & Lenssen (2019) library (version 2.0.4). The pretraining of each base-sized GRABLI model took approximately 9 hours on 4 NVIDIA V100 GPUs and 8 CPU cores. The pretraining of large-sized GRABLI models took 10 hours on 8 NVIDIA V100 GPUs and 16 CPU cores. For both base and large models we adopted ZeRO (Rajbhandari et al., 2020) stage 2 from Deepspeed Rasley et al. (2020). For all QA and linking experiments we adopted a machine with single NVIDIA V100 GPU.

