# OpenReview forum: "GRABLI: Cross-Modal Knowledge Graph Alignment for Biomedical Language Models"
_ICLR.cc/2025/Conference — Submitted to ICLR 2025_

### Official Review · Reviewer_UfGv · 2024-10-28

**Soundness:** 2
**Presentation:** 2
**Contribution:** 2
**Rating:** 3
**Confidence:** 3

**Summary:**

This paper proposes pre-training language models for the text and knowledge graph alignment. The authors evaluate the pre-trained model on several biomedical NLP tasks. The authors also compared several graph encoders and showed the importance of attention aggregation and local graph context.

**Strengths:**

- The model improves over several baseline models on several biomedical NLP tasks.
- The method is simple, and the resulting model is not task-specific and is easy to use.
- Several ablation analyses are provided.

**Weaknesses:**

- The novelty of pre-training the model using external knowledge is low. The pre-training is not always effective; for example, the model does not work well with SapBERT and GEBERT, which have already incorporated external knowledge.
- The performance is compared with existing models only on QA tasks, and the performance is lower than or comparable to the existing models. The relation extraction performance on three datasets in Table 6 shows much lower performance than existing models because the baseline models are weak. It is unclear how effective the model is for state-of-the-art models.

**Questions:**

See the weaknesses above.

---

### Official Review · Reviewer_quPA · 2024-11-01

**Soundness:** 2
**Presentation:** 3
**Contribution:** 2
**Rating:** 3
**Confidence:** 4

**Summary:**

In this work, the authors propose to train jointly a Graph Neural Network and a Masked Language Model with an auxiliary contrastive alignment objective, in order to enhance entity representations. To achieve their goal efficiently, the authors sample a local KG subgraph based on UMLS entities discovered using BERN2 in relevant documents mentioning the concept. After initializing the embeddings of concepts in the local subgraph with a LM, the MLM and GNN training are performed in parallel, and aligned using contrastive batches of size 256.

**Strengths:**

- The proposed approach does appear novel in its specific implementation. While many prior works combine GNN/MLM trainings (ERNIE, DRAGON, JAKET, ...), their approach has visible differences with the method proposed in this paper. The local subnetwork sampling is interesting, because it reduces the graph size and makes the joint training tractable.
- The comparative analysis of several GNN methods combined with MLM alignment (Table 3) is insightful, as well as the ablation study on hyperparameters (Table 4).
- Figure 1 is clear and informative.

**Weaknesses:**

- Coverage of the contemporary literature is insufficient. PubMedBERT, BioLinkBERT, and SapBERT are all models released in 2021-2022, which is more than 2 years ago. Much work has been produced on the topic since then, which is not engaged in this paper. A quick search yielded `BioLORD-2023: Semantic Textual Representations Fusing LLM and Clinical Knowledge Graph Insights`, `CODER: Knowledge-infused cross-lingual medical term embedding for term normalization` and `MedCPT: Contrastive Pre-trained Transformers with Large-scale PubMed Search Logs for Zero-shot Biomedical Information Retrieval`. Not engaging these recent works means that stronger baselines have been omitted.
- The gains over chosen baselines are insufficient. Concerning QA (Table 1), the proposed strategy does not perform as well as the DRAGON models, and the improvements produced by GRABLI training, as measured in the paper, are not significant (one would need at least 2x the standard deviation for this, while the measured improvement rarely exceeds even 1x the standard deviation). Concerning NEL (Table 2), pretrained models undergoing GRABLI training do not score better than SapBERT, and SapBERT performs worse after GRABLI training. Given that the proposed methodology is significantly more complicated than SapBERT training, obvious gains in the evaluated benchmarks would be desirable. Concerning RE (Table 6), no baseline was provided and the GRABLI training improvements are again not likely to be significant. Finally, SapBERT is not SOTA anymore and stronger baselines should have been considered.
- The complexity added by the GNN does not seem necessary either, since LLM embedding of a linearized graph performs about equally well, as per Table 3. Furthermore, it's unclear whether two different LMs are required or whether one LM could be used for text and linearized graph embedding, as in other related works.
- The quality of the semantic entity representations themselves is not evaluated, for instance through Link Prediction tasks. The article is only focusing on the training methodology aspect but does not provide sufficient evidence that the training methodology performs better or is more efficient than other training methodologies.

**Questions:**

- How does your work compare to more recent Biomedical Language Models for the considered tasks? In particular, how "brief" is your second-stage finetuning compared to SapBERT, taking into account the sucessive GNN training costs.
- Have you considered comparing the obtained performance gains against the FLOP cost of GRABLI vs other training methodologies such as SapBERT, CODER, or BioLORD?
- What is the textual input embedding used during the training of the GNN? It is mentioned in the text that it is initialized with the representation from the LLM; does that mean that this is the representation of the LLM pre-training, or is that representation continually computed anew during training using the current LLM?
- Would it be possible to reduce the standard deviation of training runs, to obtain more significant results?

---

### Official Review · Reviewer_rm6h · 2024-11-03

**Soundness:** 2
**Presentation:** 3
**Contribution:** 2
**Rating:** 5
**Confidence:** 4

**Summary:**

* This paper proposes a pre-training method that enriches an LM with external knowledge (i.e., Unified Medical Language System (UMLS) knowledge graph)
* The authors employ two encoders---text LM encoder and a GAT graph encoder. Biomedical mentions in PubMed abstracts are linked to the Unified Medical Language System KG. Then, the text mentioned, and the local KG subgraphs are used as cross-modal positive samples to align these two encoders.
* The proposed method's effectiveness is demonstrated on three downstream tasks: QA, entity linking, and relation extraction.

**Strengths:**

* The paper is well-structured and easy to follow. The proposed method is also straightforward and makes sense.
* The effectiveness of the proposed framework is demonstrated on multiple datasets (corresponding to different tasks and backbone models)
* Several analyses were conducted to show the effectiveness of different design choices (e.g., graph representation method, training loss)

**Weaknesses:**

* The empirical results are not strong enough to support the claims.
	* The proposed method underperforms task-specific models most of the time.
	* The improvements over baseline backbone models on QA are inconsistent. For example, `GRABLI (GNN)` outperforms the corresponding `PubMedBERT` and `BioLinkBERT-base` models with a large margin on `BioASQ`, but the same as `BioLinkBERT-large`. Considering the significant standard deviation, the usefulness of the proposed method on QA is less convincing. This makes the proposed method less attractive, especially since the author's main criticism of previous work is that previous `work mostly fine-tuned LMs for entity linking, limiting their applicability beyond this specific task.`
	* Based on the results in Table 4, the MLM training objective is more important than the proposed alignment training loss.
* One minor concern is reproducibility because the authors employ an existing tool (BERN2) to build their pretraining data. The authors mention that `The source code as well as pre-trained models will be released upon paper acceptance.` I hope they provide sufficient details for other researchers to reproduce their results. Also, see Q3.

**Questions:**

1. Line 421: `QA task` -> `entity linking task`
2. It would be interesting to see how the proposed method is sensitive to the quality of pretraining data (e.g., a large portion of mentions in PubMed are linked to the incorrect concepts in UMLS, which is very likely to be the case)
3. Do you use any method to avoid data leakage? i.e., making sure the pretraining data do not contain articles used in downstream evaluation datasets (e.g., PubMedQA, NCBI)

---

### Official Review · Reviewer_j6Kd · 2024-11-03

**Soundness:** 2
**Presentation:** 2
**Contribution:** 2
**Rating:** 6
**Confidence:** 3

**Summary:**

The paper presents a novel pre-training method called GRABLI (Knowledge Graph and Biomedical Language Model Alignment), designed to bridge the gap between pre-trained language models (LMs) and biomedical Knowledge Graphs (KGs). The method aims to enhance language models by integrating complex, domain-specific factual information from biomedical KGs directly into the LM's training process. This is achieved by normalizing biomedical concept mentions within texts to the Unified Medical Language System (UMLS) and using these normalized mentions to generate local KG subgraphs that serve as cross-modal positive samples. Empirical testing shows that GRABLI improves performance across various biomedical language understanding tasks.

**Strengths:**

- Originality: The GRABLI approach introduces a novel integration of KGs and LMs through cross-modal alignment, aiming to improve the semantic richness of LMs with domain-specific details from KGs. This dual-encoder strategy is distinctive in its utilization of both a graph neural network (GNN) and a pre-trained LM to create a unified representation space.

- Quality: The technical evaluation of the GRABLI framework is robust, featuring a comprehensive set of experiments across various benchmarks for question answering and entity linking tasks.

- Clarity: The paper is well-structured and articulates the research problem and the proposed solution clearly.

- Significance: The significance of GRABLI lies in its potential to enhance the performance of biomedical LMs by integrating domain-specific knowledge.

**Weaknesses:**

- When evaluating the performance of GRABLI against existing biomedical ontology encoders, the authors should provide details regarding the size of each model, specifically the number of trainable weights added when including GRABLI (e.g. the addition of the GNN). This analysis is crucial because the performance improvements observed with GRABLI may be due to an increase in model size rather than its inherent effectiveness in learning superior medical concept representations.

- Limitations: The authors should discuss the limitations of the proposed methodology, including the limited data used in the pre-training stage (1.5M sentences and 600K nodes only).

- Computational Efficiency: There is limited discussion on the computational demands of GRABLI, particularly when scaling to larger knowledge graphs or more extensive text corpora.

**Questions:**

- The code and datasets should be made publicly available for reproducibility purposes.

---

### Meta-Review · Area_Chair_NQDZ · 2024-12-16

**Metareview:**

This paper presents an approach intended to allow pretrained LMs to exploit information codified in knowledge graphs. They call this Knowledge Graph and Biomedical Language Model Alignment (GRABLI). Reviewers were in consensus that the objective proposed offers some novelty. However, ultimately the experimental setup and results were found unconvincing by the majority of reviewers.

**Additional Comments On Reviewer Discussion:**

The authors opted not to respond to reviews.

---

### Decision · Program_Chairs · 2025-01-22

Reject